

# Teaching with digital geology in the high Arctic: opportunities and challenges

Kim Senger[1], Peter Betlem[1,2], Sten-Andreas Grundvåg[3], Rafael Kenji Horota[4], Simon John Buckley[5], Aleksandra Smyrak-Sikora[1], Malte Michel Jochmann[1], Thomas Birchall[1], Julian Janocha[3], Kei Ogata[6], Lilith Kuckero[1], Rakul Maria Johannessen[7,1], Isabelle Lecomte[8], Sara Mollie Cohen[1] and Snorre Olaussen[1]

[1]Department of Arctic Geology, The University Centre in Svalbard, Longyearbyen, Norway
[2]Department of Geosciences, University of Oslo, Oslo, Norway
[3]Department of Geosciences, UiT The Arctic University of Norway, Norway
[4]Department of Applied Computing, UNISINOS - Universidade do Vale do Rio dos Sinos, Brazil
[5]NORCE Norwegian Research Centre, P.O.B 22 Nygårdstangen, NO-5838 Bergen
[6]Department of Earth Sciences, Environment and Resources, University of Naples "Federico II", Italy
[7]Department of Geosciences and Natural Resource Management, University of Copenhagen, 1250 Copenhagen, Denmark.
[8]Department of Earth Science, University of Bergen, Bergen, Norway

*Correspondence to*: Kim Senger, kim.senger@unis.no, +47 95 29 15 92

**Abstract**

The Covid-19 pandemic occurred at a time of major revolution in the geosciences – the era of digital geology. Digital outcrop models (DOMs) acquired from consumer drones, processed using user-friendly photogrammetric software and shared with the wider audience through online platforms are a cornerstone of this digital geological revolution. Integration of DOMs with other geoscientific data, such as geological maps, satellite imagery, terrain models, geophysical data and field observations strengthens their application in both research and education. Teaching geology with digital tools advances students' learning experience by providing access to spectacular outcrops, enhancing visualization of 3D geological structures and improving data integration. Similarly, active use of DOMs to integrate new field observations will facilitate more effective fieldwork and quantitative research. From a student's perspective, geo-referenced and scaled DOMs allow an improved appreciation of scale and of 3D architecture, a major threshold concept in geoscientific education.

In view of the Covid-19 pandemic, DOMs allow to bring geoscientists to the outcrops digitally. At the University Centre in Svalbard (UNIS), located at 78°N in Longyearbyen in Arctic Norway, DOMs are actively used even in non-pandemic years, as the summer field season is short and not overlapping with the Bachelor "Arctic Geology" course package held from January to June each year. In 2017, we at UNIS developed a new course ('AG222: Integrated Geological Methods: from outcrop to geomodel') to encourage the use of emerging techniques like DOMs and data integration to solve authentic geoscientific challenges. In parallel, we have established the open access Svalbox geoscientific portal, which forms the backbone of the AG222 course activities and provides easy access to a growing number of DOMs, 360° imagery, subsurface data and published geoscientific data from Svalbard. Considering the rapid onset of the Covid-19 pandemic, the Svalbox portal and the pre-Covid work on digital techniques in AG222 allowed us to rapidly adapt and fulfill at least some of the students' learning objectives during the pandemic. In this contribution, we provide an overview of the course development and share



experiences from running the AG222 course and the Svalbox platform, both before and during the Covid-19

pandemic.

**Introduction**

From 13th of March 2020 until the summer break, all university-level teaching in Norway (including Longyearbyen, where the University Centre in Svalbard (UNIS) is located) was conducted fully digitally due to the Covid-19 pandemic. In Svalbard, this occurred at the worst possible time with respect to the geology bachelor

course schedule, as the sun only returns to Longyearbyen on 8th of March after a long dark season. March and April represent the major spring field season when snowscooters can be used to access outcrops. In the years leading up to the pandemic we developed a new methods-focused Bachelor-level course at UNIS, "AG222: Integrated geological methods – from outcrop to geomodel", focusing on digital geological techniques in order to extend our field season digitally. This focus on enhancing the value of digital geological methods in education

prior to the pandemic was instrumental during the transition to digital teaching of AG222 during the pandemic.

Digital outcrop models (DOMs) have been used for several decades, particularly by the petroleum industry with its need for quantitative data on reservoir architecture (Howell et al., 2014;Marques Jr et al., 2020). Initially, most DOMs were collected by ground- or helicopter-based lidar scanners (Hodgetts, 2013;Rittersbacher et al.,

2013;Buckley et al., 2008), often requiring expensive equipment and significant processing resources, time and specialist skills. The emergence of structure-from-motion (SfM) photogrammetry (Westoby et al., 2012;e.g., Smith et al., 2016), essentially utilizing many overlapping images to construct a DOM, as led to mainstream adoption of DOMs in both teaching (e.g., Senger et al., 2020;Bond and Cawood, 2020) and research (e.g., Anell et al., 2016;Marques Jr et al., 2020;Rabbel et al., 2018). We consider this a major technology-driven revolution in the

geosciences as introduced by Buckley et al. (2019a), similar in significance to the adoption of 3D seismic acquisition that revolutionized our understanding of the subsurface (Cartwright and Huuse, 2005).

To make full use of this digital geoscience revolution, we need to re-think how geology is conducted and taught ,while maintaining focus on key skill sets required by geologists in today's society. Field-based skills acquired

while in the field are central to any geoscientist's education (Mogk and Goodwin, 2012), with digital tools allowing more efficient field work. In addition, integrating DOMs into a regional geological context using complementary data sets, and harvesting these expanding data for quantitative studies, we can take the next step towards "big data geoscience" (e.g., Guo et al., 2014;Bergen et al., 2019). Importantly, we should bring this geoscience revolution to geoscience students at an early stage, by developing skills-oriented courses where tasks are authentic to real-

life problems faced by professional geologists.

Actively participating in the digital geoscience revolution has several benefits, including improved accessibility for those that cannot participate in field work (Whitmeyer et al., 2020;Bond and Cawood, 2020), a prolonged field season (Senger and Nordmo, 2020), potential for field work preparation and thus more effective and targeted field

work and reduction in associated environmental and economic costs of field campaigns. It should, however, be stressed that geoscientific field work should not be purely digital. Participation in traditional field work and field





excursions are fundamental aspects of becoming a geoscientist (Mogk and Goodwin, 2012;Kastens et al., 2009), and digital tools should, in our opinion, complement these, rather than replace them.

Scientific literature on the application of photogrammetry in geology increases exponentially (Fig. 1), in line with technological advances. More importantly is that DOMs are readily available to the global geoscientific community through a number of open access repositories such as e-Rock (https://www.e-rock.co.uk/; global coverage; Cawood and Bond, 2019), V3Geo (https://v3geo.com/), Mosis HUB (https://mosis.vizlab.cc/xp/models), Virtual Australia (http://ausgeol.org/atlas) or Svalbox (http://svalbox.no/;

Svalbard coverage; Senger et al., 2020). All of these are useful for teaching purposes and have been heavily used during the Covid-19 pandemic, as they provide examples of a number of lithologies and structural styles that can serve as a backbone to digital teaching exercises.

An important challenge yet to be fully addressed is that while we as a geoscientific community exponentially
collect more DOMs globally, actively using them for further work is hampered by varying standards, available metadata and access regulations to the actual models. Furthermore, utilizing DOMs to their full potential requires site-specific knowledge of the regional significance of the outcrop, thus often relying on geologists with local expertise and efficiently harvesting the sheer volume of scientific knowledge about a particular area such as Svalbard (Fig. 1).

From a broader perspective, we as educators also need to consider how best to train geoscientists to exploit the digital geoscience revolution to their advantage. The benefits are clear, but the challenges with numerous software (some open-source, but most proprietary and costly) and using cross-software workflows can also be daunting. In essence, we can ask ourselves the question of how to best teach digital geosciences, and whether we can teach it
in an active and integrated fashion.

In this contribution, we share our experiences of teaching digital geosciences at UNIS, primarily related to a 15 ECTS bachelor-level course (AG222: Integrated Geological Methods: from outcrop to geomodel; ECTS = European Credit Transfer and Accumulation System; 60 ECTS = one full-time study year) offered annually since
2018 that actively uses the Svalbox geoscience platform. We outline our experience of both the course development and incremental optimization, including a fully digital field campaign organized in April 2020 during the Covid-19 pandemic. Finally, we identify knowledge gaps that should be addressed to maximize the experience from the Covid-19 pandemic to further improve geoscientific field teaching in the high Arctic.

**The Svalbox geoscience platform**
Svalbox, developed at UNIS since 2017, and introduced by Senger et al. (2020), is a platform that strives to integrate multi-physical and multi-scale geoscientific data from Svalbard for more effective teaching and research. Svalbox (Video 1 in supplementary material; Discover Svalbard's Geology with Svalbox) comprises both a public web front end sharing most of the openly accessible data (Video 2 in supplementary material; Svalbox: Introducing the Svalbox.no online map portal), and a UNIS-internal package integrating also classified data in thematic Petrel
projects (Video 3 in supplementary material; Svalbox - what is it and what data do we integrate?).





Most of the Svalbox elements can also be used by geoscience courses not run by UNIS, and our ambition is to generate high-quality datasets and educational material to bring Svalbard's exciting geological evolution to classrooms around the world.

**The AG222 course: establishment and incremental optimization**

Being based in Svalbard, an Arctic archipelago located at 74-81°N, the AG222 course had from the onset been designed with the extreme seasonal cycle in mind (Fig. 3; Fig. 4; https://youtu.be/Pjr-4L5zqE8). The "Integrated Geological Methods: from outcrop to geomodel" (AG222) course was developed at UNIS in 2017 and was run annually from 2018 onwards. In 2018 and 2019, the course was run as planned from January to late May, with up to 20 students admitted each year and a significant field component (Senger et al., 2020). In 2020, the Covid-19

pandemic led to the second half of the course being run fully digitally with students dispersed throughout Europe. Only one day of fieldwork was possible, with the main field campaign to Billefjorden having to be run virtually (Smyrak-Sikora et al., 2020). In January 2021, the course started as a fully digital course but with students in Longyearbyen. Since no Covid-19 cases were reported from Svalbard until submission of this paper in mid-March 2021, some physical teaching was implemented at UNIS in February 2021 and field excursions were run as planned

in March 2021.

The overall ambition central to the course development was to provide a new course actively using emerging digital geological techniques applied to geoscience challenges relevant to Svalbard, with the key outcome of developing the problem-solving skills required by geology graduates in their future careers, especially relevant in industry. An important component focuses on the integration of different techniques and data sources, which are

important skill sets for professional geologists that also need to act multi-disciplinary to solve real-life geological challenges. Furthermore, the course was designed to complement the existing course "The Tectonic and Sedimentary History of Svalbard" (AG209; (https://www.unis.no/course/ag-209-the-tectonic-and-sedimentary-history-of-svalbard/)) running at the same time, attended by the same students and visiting complementary field sites.

Transport to the localities is primarily by snowscooter, which often increases engagement for many students (see Supplementary Video 4; AG222 excursion @ UNIS - February 2020). The Billefjorden excursion involves a long (ca 4-5 hours) journey to the field area, relying on good visibility as it involves crossing major glaciers exposed to bad weather. Once the field area is reached a base camp is established in a hotel in Pyramiden, an abandoned coal mining settlement (Fig. 3). From Pyramiden, all localities within the entire Billefjorden Trough are easily

accessible within short driving distance (see .gpx files with localities and route in supplementary material). Geological stops are typically up to one hour long, and summaries of the main learnings from the visited localities are conducted through student presentations once back in the sheltered base camp.

**The AG222 course: case studies**

All course modules and assessments in the AG222 course are designed with a strong component on real-world

application – i.e. they should represent tasks that professional geologists working in the private or public sector may face in their future careers. There is no final exam, and the course grade reflects tasks conducted throughout





the semester, combining both group and individual assessments. In this section, we present the main course modules and associated assessments. Adequate material is provided in the supplementary material to allow implementation of these elsewhere. Naturally, the exercises are focused on Svalbard's geological record, but the
concept can easily be applied in other areas. For educators and students not familiar with Svalbard's geological evolution, Electronic appendix Table 2 provides a list of key literature.

### Data mining and integration

AG222 starts in January, in the middle of the polar night. Before the light and sun return in mid-February and early
March respectively (Fig. 4), the students familiarize themselves with the different tools (software and online resources; see Table 3) and data sets they will be using throughout the semester. In parallel, the sister course AG209 introduces Svalbard's tectono-stratigraphic evolution and main concepts such as source-to-sink.

The exercises are built around practical tasks that are routinely used by geologists working in Svalbard – including planning field campaigns in different seasons (using geological maps, satellite imagery, oblique aerial images and
DOMs), investigating what research has already been conducted in a given area (using literature and the ResearchInSvalbard database) and integrating all available databases and tools (e.g., Svalbox, SvalSIM, StoryMaps, online resources). The skills acquired through the exercises are strengthened by regular student presentations to their peers, and generation of "how-to" videos shared through the SvalDocs Wiki platform (which builds up over time) and the course page on Microsoft Teams (which was implemented from 2020 and is only
accessible for the current AG222 students). Furthermore, the rest of the AG222 course builds on the learned practical skills and actively uses these in tasks later during the semester.

### Digital models: acquisition, processing, interpretation and integration

Cost-effective georeferenced digital outcrop models (DOMs) are a breakthrough for geoscientific research and
education, and naturally are included as a central part of the AG222 course. Students learn the entire workflow from image acquisition to integration of DOMs with complementary data in the same area (Fig. 6). The dark season and snow cover prohibit own image acquisition of outcrops, but photos of everyday objects and oblique images from TopoSvalbard are used as input data. In addition, photographs acquired during the summer field season by UNIS staff are provided for generating DOMs.

DOM processing is taught using a custom-build online e-learning module (https://unisvalbard.github.io/Geo-SfM/landing-page.html) openly shared on the Github repository. Agisoft Metashape Professional is used for the SfM processing, and is also used for line interpretations in research projects at UNIS. In the AG222 course, models are exported to LIME (Buckley et al., 2019b) for interpretation and integration. In LIME, students make measurements and observations using basic lines, orientation planes, panels and points of interest (see
http://www.virtualoutcrop.com/resources/videos for details). Later, they can integrate terrain models, maps and remote sensing imagery to give regional context and appreciate the variations in scale between different datasets. They also create panel interpretations (Buckley et al., 2019) and present a characterisation of their selected DOM in group presentations. The Svalbox database (Senger et al., 2020) provides the students with an overview of the





available DOMs from Svalbard and allows them to access these for their research projects. DOMs are all available
on the Svalbox online portal ([www.svalbox.no/map](www.svalbox.no/map)) and selected ones are also uploaded to the V3Geo, including
the "flagship" DOM of Festningen ([https://v3geo.com/model/226](https://v3geo.com/model/226)).

**Virtual field trips**

Virtual field trips (VFTs) integrate numerous elements (digital outcrop models, publications, 360 imagery, photos,
geological maps, satellite imagery etc.) within a geological story line suited for a specific target audience. VFTs
can be actively presented to an audience or made accessible for individuals to follow at their own pace. Building
a VFT is as rewarding as following one, as it fosters creativity and group work. Furthermore, the oral presentation
of the VFT by the student groups simulates authentic experiences of presenting at international conferences.

A central assignment in AG222 (25% of course grade) involves the students developing a VFT to a given location
and presenting it to a wider audience (i.e. the AG222 class, AG222 guest lecturers and UNIS geology staff). The
task is conducted from the onset of the course and finalizes by the end of February when the light slowly returns
to Longyearbyen. As such, the students only use provided material and data to develop a catchy and creative VFT.
From 2018 to 2020, VFTs were organized as in-person presentations, where all elements were linked through a
standard presentation (PowerPoint or Prezi). From 2021, we have adopted the online ArcGIS StoryMaps approach
to develop the VFTs. This allows more creativity with respect to directly embedding DOMs, videos and other
central elements. As a further benefit, the resulting VFTs are permanently available on the Svalbox portal
([http://www.svalbox.no/virtual-field-trips/](http://www.svalbox.no/virtual-field-trips/)) for further exposure and to contribute to a growing VFT database from
Svalbard. The supplementary material provides the key public-domain resources and tools required to design
Svalbard VFT experience.


**Synthetic seismic**

Seismic modelling allows the direct correlation of outcrops with seismic data, aiding to quantify what geological
features are visible on the latter. By creating geomodels from digital outcrop models, students also truly appreciate
several factors that control seismic images. Building geomodels involves conducting line interpretations on digital
outcrops to establish the structures, then assigning elastic properties (P- and S-wave velocity and density) to these
from literature or borehole data crossing the same stratigraphic interval of interest. Once both structures and elastic
properties are in place, seismic modelling is applied (e.g., Rabbel et al., 2018;Anell et al., 2016). Noise is usually
also added for more realistic results (Lubrano-Lavadera et al., 2019). The synthetic seismic profiles are then
overlain on the DOM and class discussions focus on understanding what geological features are discernible, and
how seismic acquisition parameters (primarily frequency and illumination angle) affect seismic imaging.

In AG222, the Triassic succession at Kvalpynten in south-western Edgeøya (https://v3geo.com/model/90; Anell
et al., 2016;Smyrak-Sikora et al., 2020) was primarily used from 2018 to 2020 as an excellent case study. The
"seismic-scale" outcrop displays two contrasting geological features. The lower part of the outcrop is dominated
by growth-faults with small half-grabens infilled by siliciclastic syn-sedimentary deposits. In the upper part, very
low angle clinoforms related to the progradation of the world's largest delta plain in the Triassic (Klausen et al.,
2019) are barely apparent even at the ca. 7 km long outcrop.





### Billefjorden claim application

The license claim application is an intensive group assessment worth 30% of the AG222 grade, conducted in an
intensive 3.5 week period building around the main 4-day AG222 field excursion to the Billefjorden Trough. The
application follows a strict template, as do authentic license applications, and challenges the students to compile a
convincing overview of the petroleum system elements to secure a fictive claim in the field area (Fig. 8).
Furthermore, an exact drilling location and well prognosis to 1300-1600 m depth must be provided, along with
sub-surface correlations and environmental considerations relating to petroleum exploration of this sensitive area.

This assignment is by far the most authentic of all AG222 tasks, as the only oil discovery in Svalbard was reported
from the area, a consequence of coal exploration by the Russian company Trust Arktikugol in the 1990s (Senger
et al., 2019). Indeed, there were concrete plans as late as 2004 to drill a serious petroleum exploration borehole in
the area (Senger et al., 2019). Obviously, these plans never materialized, but the AG222 students can experience
this authenticity, and make full use of their geological understanding to compete between the groups for the best
overall license claim application.

The students integrate pre-existing material to learn as much as they can about the Billefjorden Trough prior to the
field excursion, including exposure to DOMs from summer field work and a comprehensive Petrel project of the
basin (including wells, published cross-sections, digital terrain model, satellite imagery, geological maps). During
the field campaign, students get both a basin-scale exposure at overview stops, but also collect samples and
information (structural and sedimentological data) to be used in their application. The digital field notebook
(Senger and Nordmo, 2020) is used to organize each group's field data, and the resulting FieldMove project is a
compulsory appendix to the license claim application.

### Near-town geology: drill core & outcrop sedimentology and structures

A core shed near UNIS stores more than 60 km of drill cores collected by the local mining company SNSK for
coal exploration, and for scientific purposes (e.g., UNIS $CO_2$ lab; Olaussen et al., 2019). The stratigraphy covers
the successions outcropping in the mountain sides near Longyearbyen. Since geologists from the mining company
have contributed to teaching at UNIS for almost two decades, it was natural to make this unique material available
for student learning. Accompanied by the company-geologist, students visit the shed to get first-hand knowledge
about diamond drilling in the high Arctic, and sedimentary drill core logging. They practice detailed logging of
cores and logging under time pressure, role-playing that bad weather is coming and the helicopter waiting to pick
them up, the latter often the case for real Arctic drill site geologists. These exercises help them to understand the
geology of the surrounding mountain, and they build the basis for later field-logging exercises, be it outcrop scale
or making rough logs of mountain sides from the distance.

By early May, the snow begins to melt and outcrops near Longyearbyen allow for conducting some meaningful
fieldwork along one of Svalbard's arguably best exposures: the c. 2 km continuous outcrop transect between
Longyearbyen and the airport. The transect excellently exposes a succession of alternating sandstones, siltstones
and shales of Early Cretaceous age, thus enabling high-resolution bed-to-bed scale investigations, as well as lateral





tracing and mapping of depositional elements. In addition, c. 4.5 km of drill cores are available from parts of the

Mesozoic succession drilled by the UNIS $CO_2$ lab project (Olaussen et al., 2019). These cores penetrate the same
succession that is exposed in the Flyplassveien outcrop. The combination of outcrops and drill cores allow for a
detailed and integrated sedimentological and structural characterization of the investigated succession. The
students focus on and practice various methods for acquiring and presenting sedimentological and structural data.
Structural data are for example collected both with a traditional compass and by using digital tools like tablets and

smartphones (Novakova and Pavlis, 2017). In recent years, selected drill cores are digitized using SfM
photogrammetry (Betlem et al., 2020b) and shared on Svalbox.

The collected scanline data are discussed in regard to the mechanical stratigraphy of the succession, particularly
focusing on how bed thickness and lithology correlate with fracture intensity. In addition, field data are integrated
with DOM data to extend the area of investigation to include the inaccessible parts of the outcrop. This also

increases the length of the field season that is notoriously short in the high Arctic.

**Paleoclimate drilling poster presentation**

Svalbard's geological record provides a unique window into deep-time paleoclimatic events of global significance
(Senger et al., 2021). The Permian-Triassic boundary (P-Tr; Zuchuat et al., 2020) and the Paleocene-Eocene

Thermal Maximum (PETM; Dypvik et al., 2011) are just two examples of globally significant events preserved in
Svalbard's rock record and studied in detail in drill cores from Svalbard. The P-Tr boundary was targeted by the
last drilling in Svalbard, with two ca. 100 m deep research boreholes drilled and fully cored at Deltadalen in 2014
(Zuchuat et al., 2020).

The AG222 students finalize the course with an individual poster presentation that presents a "Deltadalen-style"

drilling proposal for a 100-200 m deep paleoclimate research borehole to target an assigned interval of interest
(Snowball Earth; end Permian mass extinction; Early Cretaceous oceanic anoxic events; PETM). As with the
petroleum drilling in Billefjorden, the students need to utilize all their skill sets to find a suitable location, and
propose a realistic concrete target including a well prognosis. The assignment is individual, and its presentation at
the final day of the AG222 course provides an authentic experience in presenting posters at scientific conferences.

**Discussion**

**Digital outcrop models – a game changer for digital teaching**

DOMs are in our opinion a cornerstone of the ongoing geoscience revolution and a game changer for digital
geoscience teaching methods (Fig. 9). DOMs are multi-scaled features, and thus allow the easy appreciation of
resolution (i.e., pixel resolution, in other words the size of the smallest discernible objects), scale (i.e., size of the

DOM) and perspective (i.e., viewing angle and exaggeration). DOMs can be generated across all scales, from
seismic-scale outcrops to high-resolution drill core or hand sample models, and facilitate quantitative geology,
including unprecedented possibilities for making realistic outcrop-based geological models (e.g., Larssen et al.,
2020). Integration of DOMs with shallow geophysical data, e.g Ground-Penetrating Radar (GPR), also opens up
to "see" geology beyond the outcrop, as illustrated with the paleokarst at Fortet (Supplementary video 5; The

Billefjorden Trough STOP 6- update) (Janocha et al., 2020). Along with the explosion in cost-effective DOM



acquisition from drones, the ease of sharing them with the global geoscience community through a multitude of 3D platforms (e.g. Sketchfab or V3Geo) and rendering libraries (e.g. potree, Unity) truly opens up for global digital geology teaching.

DOMs complement traditional field data collection by facilitating data acquisition in inaccessible areas, provide greater structural data sampling and reduce time spent in the field (Nesbit et al., 2020;Pringle et al., 2006;McCaffrey et al., 2010). Furthermore, DOMs are ideal for training and teaching geology, as they allow appreciation of structures from different perspectives and vertical exaggerations, student-teacher discussions in a controlled indoor environment and (digital) accessibility to the field irrespective of the participants field experience, economic and cultural background.

DOMs, particularly when derived from drone-based photographs, make inaccessible outcrops safely accessible without the risk of rock fall, avalanches, climate issues, steep and rocky terrains or wildlife (e.g., polar bears, rattlesnakes). Precisely acquired DOMs allow geologists to extract and present quantitative and qualitative geological information and detailed measurements without the need to directly access them (Larssen et al., 2020;Marques Jr et al., 2020;Senger et al., 2015a;Nesbit et al., 2020;Nesbit et al., 2018). This approach increases
the areas from which measurements can be made, which means that more statistical information can be collected, increasing the sample size and therefore reducing errors in statistical analysis (Fabuel-Perez et al., 2010;García-Sellés et al., 2011;Hodgetts, 2013).

In the same way, new attributes can be generated to highlight subtle features, helping in the interpretation by providing the basis for automated mapping approaches (McCaffrey et al., 2005;McCaffrey et al., 2010). These
either reduce the time needed for fieldwork, or make fieldwork more efficient with more data acquired over the same time interval. For comparison, Ogata et al. (2014) present > 9000 structural measurements of fractures collected on a sandstone in Svalbard over a 1-month field period, while recently acquired DOMs in the same area would exponentially reduce the time needed to collect the same data. The reduced cost of fieldwork by active use DOMs and VFTs is also considerable, both in industry and academia. This is especially relevant when entire teams
should investigate outcrops together, and discuss while observing the outcrop. However, DOMs are still not a replacement for traditional field trips, but a tool that can improve the field experience (Hodgetts, 2013), making it an efficient way to integrate and visualize multi-scalar surface and sub-surface rock data in desktop applications. The increased use of immersive Virtual Reality (iVR) already provides authentic digital field experiences (Gonzaga et al., 2018), and only the resolution and spatial limits of individual DOMs set the boundaries of what
is possible.

**Virtual reality: current and future perspectives**

Virtual Reality (VR) is a computer-generated environment that projects 3D objects and scenes that appear to be real, making the user feel like they are immersed in their surroundings. This environment is usually perceived
through a VR headset (e.g. Oculus, HTC Vive or cost-effective Google Cardboard-type headsets). In the entertainment industry VR allowed users to immerse themselves in video games and movies in characters point of view (Wu et al., 2021). VR has also been used for professional training, for instance to learn how to perform surgery or improve the quality of sports training to maximize performance (Ali et al., 2017;Pulijala et al., 2018).



In the field of geoscience education and research, VR has been relatively well explored in recent years (e.g., Zhao
et al., 2017;Marques Jr et al., 2020;Minocha et al., 2018;Horota et al., in review). DOMs have facilitated computer
rendering of scenes of field learning environments for geoscientists to analyze rock exposures digitally, and all
Svalbox models are VR-ready through SketchFab. 360° videos from the AG222 field excursions can also be
directly seen in VR. VR has been used since the onset in AG222, mostly to demonstrate the feasibility of off-the-
shelf tools like Google Earth VR, Sketchfab and the MOSIS Suite (https://mosis.vizlab.cc/). For the near future
(3-5 years) these software platforms are showing signs to move towards multi-user online access, which we believe
soon we will begin to provide iVR interaction of an entire class (ca 20 students and 5 teachers at UNIS) co-working
over the same DOM. This will allow the geoscience community to take one step closer to facilitate virtual field
access, with implications both for industry and academia. Clearly, the impact of such emerging technologies on
geoscientific training requires further research efforts.


### Course development & integration with Svalbox portal

The AG222 course was developed in parallel with the Svalbox portal, and this synergy will be optimized also in
the coming years (Fig. 10). The skills-based course requires relevant and authentic data sets for the authentic
experiences, which is provided through Svalbox. On the other hand, the AG222 course provides content to the
Svalbox portal, in particular virtual field trips developed by both staff and students. Similarly, UNIS-affiliated
research projects including MSc and PhD students contribute data sets to Svalbox, in particular DOMs and 360°
images. Over time, we envision that this will lead to an exponential increase in DOMs from Svalbard openly
available for the global geoscience community.

It must be noted that the AG222 course is inspired by state-of-the-art training offered in the petroleum industry,
with expert teams working together to solve authentic "real-world" problems. It is thus imperative that the skills
the students acquire as part of the course are applicable in the students' future careers irrespective of sector. In
addition to the technical skills learnt during the course, extra skills such as data management, group work and
handling intensive periods with heavy workloads are important elements to make the AG222 course as authentic
as possible.


### Field-based training for the petroleum industry and communication to the broader society

We can regard Svalbard as the exposed part of the subsurface of the Barents Sea, with ongoing petroleum
exploration and production. UNIS has over the past decade run excursions for the oil industry, particularly to
localities exhibiting Carboniferous to Lower Cretaceous strata. Those strata are linked to the proven reservoirs and
source rocks in the efficient petroleum systems in the southwestern Barents Sea petroleum province. The main
purpose of these field-based educational expeditions for the industry is to train the geological and geophysical staff
in the regional overview of the basins, tectonism, architecture and scale of reservoirs. The excursion/field trip is
run by a medium sized ship capable of accommodating 20 passengers. Although geological guides are handed out,
presentations in the evening of what to see the next day and repeated next morning it is a challenge to present the
localities as relevant for the normal "work station scale" (e.g. Petrel, Landmark etc.) at the office. However, if the
participants upfront on their own computer play with a DOM of the specific outcrop to be visited it will be possible





to better understand the scale and architecture of the geology to be visited. We thus foresee enhanced use of Svalbox in such targeted field campaigns, particularly when the Covid-19 pandemic passes and such excursions once again become feasible.


**Covid-19 implications**

On the 13th of March 2020, the Covid-19 pandemic forced the cancellation of all fieldwork at UNIS. 17 bachelor students were able to continue remotely as courses running at the time were continued digitally. The original plan included a four-day long snowmobile fieldtrip to the Carboniferous rift basin located in central Spitsbergen. The

structure of the excursion was kept as close to the original plan as possible (Fig.11). Instead of the work at a real outcrop, students were tasked with preparing "digital geological stops" of assigned sites throughout the basin and present these to the entire class through a publicly available video (Green box in Fig. 11; See the playlist here: https://www.youtube.com/watch?v=_Izk4yhEN2Y&list=PLaERIU24EpWf93UbEB701vFTqwBPG6CpY). The videos included DOMs from Svalbox, geological maps, aerial images, Google Earth overviews, georeferenced

photos and measurements and notes from FieldMove projects compiled by students taking the course in previous years. Following the presentation of these field guides, additional information and discussions were facilitated by the lecturers. The students, in groups of 3-4, were tasked to identify potential hydrocarbon prospects, and apply for a fictional claim application in Billefjorden, as described above.

The qualitative feedback collected from the students and teachers clearly points out that the virtual field excursions

cannot replace real field experience. Principal geological tests such as Mohs hardness tests, grain/crystal size analysis, or ground-truthing structural orientation measurements, which are a foundation of bachelor-level courses, cannot be performed virtually. Virtual field excursions can, however, contribute to the field-based education and serve as an introduction to the study area and function as a substitute for snow-covered or inaccessible localities or when a planned field excursion needs to be adapted to harsh weather conditions. Ultimately, our experience

suggests that there are real benefits to virtual excursions only if it is combined with real field work of the same or comparable geology.

The sister course AG-209 was also forced to adapt to the Covid-19 situation. Term projects based on field research were changed to literature research term projects, and the final assessment was changed from a graded written exam to pass/fail with an oral exam. In addition, the learning outcomes had to be changed, mainly due to the fact

that education of Arctic field geologists is impossible online, and the few field days students have had by then were only introductory. Logging and interpretation exercises, based on photographs of drill cores and outcrops instead of real rocks, were performed with surprisingly good outcome. This result would however not have been possible without the thorough introduction to sedimentary logging and describing of rocks which had been done in field and classroom prior to the Covid-19 lockdown and the students' prior skills.


**Future perspectives**

The AG222 course will continue to be offered every year and thus allows sustainable and incremental optimization through integrating emerging tools. At the moment, we are developing open access online modules for all the



course modules, inspired by the successful Geo-SfM course module (Table 1). In addition, we are continuously

testing new tools, for instance smartphones and tablet with in-built Lidar scanners (e.g., iPhone 12 Pro, 3D printers, VR technologies, thermal cameras, drone-mounted sensors) to push the boundaries of digital geological techniques. Perhaps more importantly than testing and sharing experiences from new hardware are the efforts to outline best practice documents for the many important cross-software workflows, along the from outcrop to geomodel framework.

Our overall vision is that the Svalbox platform will facilitate free and easy access to all the collected data elements. This innovative approach building on FAIR (i.e. findable, accessible, inter-operable and re-usable) data principles (Mons et al., 2017) and the open data movement will exponentially enhance the use of Svalbox DOMs beyond UNIS and contribute to squeezing out more information from already collected data. The ongoing digitization efforts of the subsurface (e.g., Nguyen et al., 2020), of vital importance for many geo-energy applications (e.g.

petroleum exploration & production, $CO_2$ storage, geothermal energy, gas and nuclear waste storage), will be able to use DOMs from a range of lithologies to test and train algorithms to facilitate the (semi-)automatic interpretation of the outcrops, including machine learning and big data analyses. Furthermore, geoscientists will ideally be able to put together all Svalbox elements into thematic virtual field trips at a click of a button. Similarly, educators at UNIS and beyond can already now use Svalbox elements to generate online and class-based course modules. As

an example, UNIS staff is currently involved in developing two course modules, "Deep-time paleoclimate in the Svalbard rock record" and "Petroleum systems of Svalbard", to be offered also to students not physically in Svalbard.

**Summary and conclusion**

In this contribution, we have outlined a Bachelor-level course on integrated geological methods developed at the

world's northernmost university in Longyearbyen, Svalbard. The focus on digital tools, and in particular digital outcrops, not only extends the short Arctic field season, but also facilitated running the second half of the course fully digitally during the global Covid-19 pandemic in 2020.

We have provided an overview of the main course elements. We conclude that the digital geoscience revolution is among us and that we as educators need to embrace it – not to replace traditional fieldwork, but to complement it

and exploit the synergies. There is no better place in the world than Svalbard to do this – as digital geology also significantly enhances our field season, and the geology of Svalbard is truly a playground for any geologist. The Svalbox portal is our contribution to open up this playground to the global geoscientific community.

**Acknowledgements**

The AG222 course is fully financed by UNIS, with significant course and Svalbox development costs financed

through numerous co-operation grants from the University of the Arctic (UArctic). Digital outcrop models freely available on Svalbox are acquired using both UNIS internal funds and external projects, notably the Research Centre for Arctic Petroleum Exploration (ARCEx), the Norwegian CCS center (NCCS), the Suprabasins project led by the University of Oslo and the Petroleum Research School of Norway (NfiP). The iEarth Centre for Integrated Earth Science Education provided seed funds to develop a virtual field trip to Festningen. The VOG



Group at NORCE added a selection of Svalbox models to the V3Geo portal. We sincerely appreciate all feedback from UNIS colleagues and data sharing from MSc and PhD students and – of course – all the students of the AG222 course at UNIS over the years.

**Data availability**

Digital outcrop models available via [www.svalbox.no](www.svalbox.no), and Svalbox Petrel projects available from Kim Senger
for non-commercial projects.

**Code availability**

Available via Svalbox GitHub repository, [https://github.com/svalbox?language=python](https://github.com/svalbox?language=python)

**Author contribution**

Conceptualization - KS

Data curation – PB, TB, JJ, KS

Funding acquisition – KS, IL, SB, SO

Software – PB, SB

Visualization – KS, RKH, AS, TB, JJ, LK, RMJ

Writing – original draft preparation - KS

Writing – review & editing – KS, PB, SAG, RKH, SJB, AS, MMJ, TB, JJ, KO, LK, RMJ, IL, SMC, SO

**Competing interests**

The authors declare that they have no conflict of interest.

**Tables**

**Table 1**: Overview of central elements collectively comprising Svalbox elements.

| Main element/Sub-element | Purpose | Data type | Accessibility | Reference/link |
|---|---|---|---|---|
| **SvalDocs** | Provide accessible documentation on how to use Svalbox, and a platform to share documents generated through Svalbox | | | |
| Workflows | Best practice for software, data acquisition | Documentation | Internally at UNIS | |
| Teaching material/e-learning | Relevant exercises and full course modules | Course material | Openly accessible | [https://unisvalbard.github.io/Geo-SfM/landing-page.html](https://unisvalbard.github.io/Geo-SfM/landing-page.html) |



| | | | | |
|---|---|---|---|---|
| Virtual field trips | Storytelling based on Svalbox data elements | Stories | Openly accessible | http://www.svalbox.no/virtual-field-trips/ |
| Literature | Dynamically updated list of literature included in Svalbox | Articles, theses, documents | Openly accessible | http://www.svalbox.no/bibliography/ |
| Case studies | Actively use Svalbox in research projects | Articles, theses | Openly accessible | Published case studies (Janocha et al., 2020;Larssen et al., 2020), compiled on http://www.svalbox.no/publications/ |
| Conference presentations and webinars | Promote and market Svalbox and its applications | Presentations, webinars | Openly accessible | Example from iEarth digital forum: https://iearth.no/en/2020/06/19/iearth-digital-learning-forum-svalbox/ |
| Videos (Svalbox YouTube channel) | Promote and market Svalbox and its applications | Videos | Openly accessible | https://www.youtube.com/channel/UCQ7tTHrKaKSBB7fxUpnabeQ |
| **Fileserver (SvalFiles)** | Robust storage of all Svalbox data | | | |
| **Acquired data** | Systematically acquire DOMs of all key outcrops in Svalbard | Photographs and processed DOMs | Openly accessible | http://www.svalbox.no/outcrops/ |
| | Sample and drill core models | Photographs and processed models | Openly accessible | Betlem et al. (2020b) |
| | Provide overview imagery from drone or handheld 360 cameras | 360 degree imagery (photos and videos) | Openly accessible | http://www.svalbox.no/map/ https://www.youtube.com/watch?v=w1XHoM1BlCM&feature=youtu.be |
| | Acquire shallow geophysics | ERT and GPR, including integrated with DOMs | Internally at UNIS | Janocha et al (2020) |
| | Documentation of field campaigns | FieldMove projects, GPX tracks | Internally at UNIS | GPX tracks on SvalGIS |
| **Integrated data** | Place DOMs and own observations in a regional perspective | Borehole data | Internally at UNIS, borehole locations openly accessible via website | Petroleum and UNIS $CO_2$ lab research boreholes (Senger et al., 2019;Olaussen et al., 2019) |
| | | Regional terrain, topography and satellite data | Streamed from NPI, openly accessible | https://geodata.npolar.no/ |
| | | Publications, including GeoTiffs, profiles and interpreted seismic | Internally at UNIS | Dallmann (2015), growing list of included publications on Svalbox website (http://www.svalbox.no/bibliography/) |
| | | Seismic, EM | Internally at UNIS, profile locations openly | e.g., Beka et al. (2017) |





| | | | |
|---|---|---|---|
| | Sedimentary logs | accessible via website Internally at UNIS, log locations openly accessible via website | www.svalbox.no/map |
| **GIS server (SvalGIS)** | Sharing of georeferenced data and metadata internally and externally through Svalbox.no | | |
| | DOMs | Openly accessible | www.svalbox.no/map |
| | 360 imagery | Openly accessible | www.svalbox.no/map |
| | Borehole locations | Openly accessible | www.svalbox.no/map |
| | Geophysical profiles | Openly accessible | www.svalbox.no/map |
| | DOMs on external repositories | Open access on partner repositories | https://v3geo.com/model/226 |

**Table 2**: Overview of the AG222 course modules.

| Course module | | Overall learning outcomes | Data and tools | Assessments in AG222 |
|---|---|---|---|---|
| 1 | Data mining and integration | Learn to use Svalbox portal, and find relevant data sets Document and use workflows | Svalbox, online resources, SvalSIM, Petrel | Virtual field trip Practical exercises |
| 2 | Digital models | Learn to acquire, process and interpret digital models | Agisoft Metashape, LIME, Svalbox and V3Geo | Practical exercises Virtual field trip License claim application |
| 3 | Wireline logs, geophysics & synthetic seismic | Learn what different geophysical and well log methods are sensitive to Bridge the gap from outcrop to geophysics through seismic modelling | SeisRoX, Petrel | Practical exercises |
| 4 | Mechanical stratigraphy | Learn to collect own structural and sedimentological data from drill cores and outcrops | Fieldwork, cores, Svalbox | Scientific poster presented in a seminar |


**Table 3**: Overview of the key assessments in AG222

| Course assessment | | Purpose | Tools used | Group/ Individual | Contribution to course grade and grading |
|---|---|---|---|---|---|
| 1 | Practical exercises | Learning skills by actively learning Keeping track of activity Use and document workflows | Svalbox, qGIS, ArcGIS, SvalSIM, StoryMaps, LIME, Petrel, online resources | I | 20% pass/fail |
| 2 | Virtual field trip | Build – in a team of experts – a virtual field trip to an assigned locality with an enchanting storyline Be creative, innovative and get to know your group | Primarily StoryMaps, with components from assessment 1 | G | 25% A-F |





| | | | | | | |
|---|---|---|---|---|---|---|
| 3 | License claim application | Play the "oil game" and find the best place to drill for petroleum in Billefjorden<br>Maximize your field experience by collecting own data to complement pre-field work digital work | Petrel, FieldMove, field geology | G | 30%<br>A-F |
| 4 | Scientific poster | How did climate change in the geological past?<br>Where in Svalbard would you drill to conduct further paleoclimatic research? | Components from assessment 1 | I | 25%<br>A-F |

**Table 4**: Overview of key resources, datasets and software used in the AG222 course

| Resource | Course module | Accessibility | Used in Covid-19-related digital teaching? | Reference/Source |
|---|---|---|---|---|
| Svalbox online portal | Data mining & integration | Anywhere with internet | Yes | Senger et al. (2020) |
| Svalbox Petrel projects | Data mining & integration | UNIS PC | Used in physical teaching by students, but only by lecturers in digital teaching | Senger et al. (2020) |
| Svalbox GIS projects (ArcGIS and QGIS) | Data mining & integration | UNIS network | Yes | Senger et al. (2020)<br>www.svalbox.no/map |
| Online geospatial resources | Data mining & integration | Anywhere with internet | Yes | https://toposvalbard.npolar.no/<br>https://geokart.npolar.no/Html5Viewer/<br>index.html?viewer=Svalbardkartet<br>https://researchinsvalbard.no/<br>http://www.svalbox.no/<br>https://factmaps.npd.no/factmaps/3_0/<br>https://geodata.npolar.no/ |
| SvalSIM | Data mining & integration | Anywhere | Yes, in both years | Saether et al. (2004) |
| Agisoft Metashape | Digital models | UNIS PC | Yes, the session was held in February | Janocha et al. (2020) |
| LIME | Digital models | Anywhere | Yes, the LIME session was held in February. In person in 2020, and hybrid (guest lecturer digital, students in person) in 2021 | Buckley et al. (2019b) |
| e-learning modules | Digital models | Anywhere with internet | Yes, in 2021 | https://unisvalbard.github.io/Geo-SfM/landing-page.html<br>Betlem et al. (2020a) |
| Smartphone/ iPad apps | | Anywhere | Only for initial part | http://www.svalbox.no/software-apps/ |
| Digital field notebook | | Anywhere | No (but data collected in 2019 was provided) | Senger and Nordmo (2020) |

**Table 5**: Examples of virtual field trips (VFTs) available on Svalbox

| VFT title | Purpose | URL |
|---|---|---|
| Repository of virtual field trips | Access point for VFTs on the Svalbox portal | http://www.svalbox.no/virtual-field-trips/ |
| Geology of Svalbard | Main landing page for Svalbox VFTs/Journeys | https://storymaps.arcgis.com/stories/<br>36cf2935a6754422bba794edeea05b9f |



| Outcrop of the week - Festningen | Short teacher-provided VFT to familiarize students with StoryMaps features | https://storymaps.arcgis.com/stories/bb3fa994b60d44a9b1312e6c2784957c |
| Discovering the fossilized world of Festningen | Teacher-provided example of a longer VFT | https://storymaps.arcgis.com/stories/5efc4f9559c348f796e643b965a5b5e9 |

**Electronic supplements**

Videos:

AG222 course overview: https://www.youtube.com/watch?v=Pjr-4L5zqE8

Svalbox overview video: (Discover Svalbard's Geology with Svalbox)

360 degree video from 2020 Botneheia excursion (Rafael; AG222 excursion @ UNIS - February 2020)

GPS tracks of AG222 excursions in 2018 and 2019 (.gpx tracks)

Digital outcrop models available via www.svalbox.no/map, some available via www.V3Geo.com


Electronic Appendix Supplementary Table 1: Key parameters of the presented AG222 course

| | |
|---|---|
| **Course requirements:** | 60 ECTS within general natural science, of which 30 ECTS within the field of geology/geosciences. Enrollment in a programme at Bachelor level. |
| **Academic content:** | The geological history of the Svalbard archipelago is a story of how tectonic and climatic processes have affected sedimentation since the Caledonian orogeny, and serves as a "window" to the Barents shelf hydrocarbon province to the south. The sparsely vegetated, well exposed and in places well-studied outcrops provide a unique opportunity for entry-level geologists to get an understanding of how geological field data are collected in the field and analysed in the office. In addition, geophysical data are integrated to enhance the holistic understanding of a particular area. Authenticity is stressed throughout the course, with practical problems to solve resolving the numerous fields requiring the robust characterization of the subsurface, including coal mining, geological $CO_2$ storage, hydrocarbon exploration, underground gas storage, geothermal exploitation, ore exploration etc. |
| **Learning outcomes:** | Upon completing the course, the students will be able to conduct focused geological field data collection in small groups, be familiar and use a broad range of geological and geophysical methods, and actively use these data to produce a realistic geological model of the subsurface.<br>**Knowledge**<br>*Upon completing the course, the students will:*<br>• develop a basic understanding of geological field mapping techniques (e.g., stratigraphic and structural mapping at outcrop and core scale)<br>• develop a basic understanding of geophysical data interpretation techniques (e.g., seismic, electric methods, wireline log interpretation)<br>• actively use modern tools (e.g., photogrammetry to construct virtual outcrops, industry-standard software for both integration and seismic modelling) to link geology and geophysics together<br>• be introduced to emerging technologies relevant for geological fieldwork, including digital outcrops, virtual reality and integration of various data.<br>**Skills**<br>*Upon completing the course, the students will be able to:*<br>• be able to work together to solve realistic and authentic subsurface characterization problems<br>• improve the understanding of the geology of an area by collecting relevant new data in the field and integrating it with pre-existing information and present their findings to the class<br>• get an authentic experience of how subsurface characterization is conducted in practice, where the key uncertainties lie and how relevant geological know-how can directly or indirectly improve the geomodel.<br><br>**General competences**<br>*Upon completing the course, the students will:*<br>• gain first-hand experience of actively working both individually and in small groups |





| | |
|---|---|
| | • improve the presentation skills by presenting their work to their peers and creatively tackling the set problems. |
| **Learning activities:** | The course will be very practical oriented, with a relatively small number of introduction and overview lectures complemented by practical exercises carried out by the students both individually and in small groups. These exercises will focus on sedimentology and structural analysis (of cores and near-town outcrops), geophysics (seismic and non-seismic interpretation), well log interpretation, geomodelling and data integration. Students will participate in a whole class excursion in Svalbard where each group will be presenting a selected geological field site to their peers.<br><br>**Total lecture hours:** 16 hours<br>**Total practical exercises/PC lab work:** 60 hours<br>**Fieldwork/excursions:** ca 3 days with overnight stay, up to 3 day-excursions<br>**Course length**: 20 weeks |
| **Course assessment:** | All compulsory learning activities (i.e. Excursions and group field work) must be approved in order to be registered for the final assessments. |

| Assessment method: | Percentage of final grade: |
|---|---|
| Practical exercises (individual work) | 20% |
| Digital field report from excursion (group work) | 30% |
| Presentation of virtual field trip (group work) | 25% |
| Presentation of scientific poster (individual work) | 25% |

| | |
|---|---|
| **Course costs for students:** | No tuition fee<br>Semester fee of ca 500 NOK<br>Contribution to food on overnight stays (200 NOK/day, max 4 days) |
| **Course costs for UNIS:** | Ca. 440 000 NOK per year, excluding salary of UNIS staff |

Electronic Appendix Supplementary Table 2: Key literature on Svalbard's geological evolution and main thematic topics.

| Main theme | Selected key references |
|---|---|
| Overall introduction to Svalbard's geology | Worsley (2008), chapter 6-8 in Dallmann (2015) |
| $CO_2$ storage efforts in Svalbard | (Olaussen et al., 2019;Braathen et al., 2012;Senger et al., 2015b;Mørk, 2013) |
| Petroleum exploration | Nøttvedt et al. (1993), Senger et al. (2019) |
| Coal exploration and production | Senger et al. (2019), chapter 11 in Dallmann (2015), Harland and Anderson (1997) |
| Deep-time paleoclimate in Svalbard's geological record | Dypvik et al. (2011) - PETM<br>Harding et al. (2011) - PETM<br>Greenwood et al. (2010) - Eocene Arctic rainforest<br>Uhl et al. (2007) - Fossil leaves in the Eocene of Spitsbergen<br>Spielhagen and Tripati (2009) - Paleocene climate fluctuations<br>Vickers et al. (2016) - Early Cretaceous climate<br>Midtkandal et al. (2016) - Aptian global anoxia<br>Hurum et al. (2016) - Barremian dinosaurs and climate<br>Jelby et al. (2020) - Jr-Cr boundary and isotope signals<br>Koevoets et al. (2016) - Jurassic isotope excursions<br>Klausen et al. (2020) - Late Triassic delta and dinosaurs<br>(Pott, 2012) - Late Triassic paleo-flora<br>Paterson and Mangerud (2020) Mid-Late Triassic palynology and climate<br>Wignall et al. (2016) - Early Triassic anoxia<br>Zuchuat et al. (2020) - PT boundary<br>Bond et al. (2015) - Mid Permian mass extinction<br>Blomeier et al. (2011) - Permian change from warm/arid to cool climate<br>Hanken and Nielsen (2013) - L.Carb.-E.Perm carbonate build-ups and climate variations<br>Hüneke et al. (2001) - L.Carb.-E.Perm carbonates and climate variations<br>Blomeier et al. (2009) - L.Carb. carbonates and Gondwana eustatic cycles<br>Berry and Marshall (2015) - Devonian forest<br>Fairchild et al. (2016) - Late Proterozoic glacial carbonate<br>Knoll and Swett (1987) – Pre-Cambrian to Cambrian transition<br>Hambrey (1982) - Late Precambrian tillites |





| | |
|---|---|
| | Bjørnerud (2010) - Kapp Lyell tillite (Neoproterozoic) |
| Paleogene | Dallmann (2015), chapter 6.10 |
| | Helland-Hansen and Grundvåg (2021) |
| Cretaceous | Dallmann (2015), chapter 6.9 |
| | Grundvåg et al. (2019) |
| Early Cretaceous magmatism | Senger et al. (2014) |
| Jurassic | Dallmann (2015), chapter 6.8 |
| | Koevoets et al. (2019) |
| | Rismyhr et al. (2018) |
| Triassic | Dallmann (2015), chapter 6.7 |
| | Anell et al. (2014) |
| | Lord et al. (2017) |
| Permian | Dallmann (2015), chapter 6.6 |
| | Blomeier et al. (2013) |
| | Sorento et al. (2020) |
| | Matysik et al. (2018) |
| Carboniferous | Dallman (2015), chapter 6.5 |
| | Smyrak-Sikora et al. (2019) |
| | Ahlborn and Stemmerik (2015) |
| Devonian | Dallmann (2015), chapter 6.4 |
| Pre-Devonian | Dallmann (2015), chapters 6.2 and 6.3 |

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



**Figures and figure captions:**

**Figure 1**: The digital geology revolution, as illustrated by the exponential growth in publications from 1990 to 2020 that include "photogrammetry" and "geology". Similarly, a marked increase is seen in publications including "geology" and "Svalbard". Data source: Google Scholar.

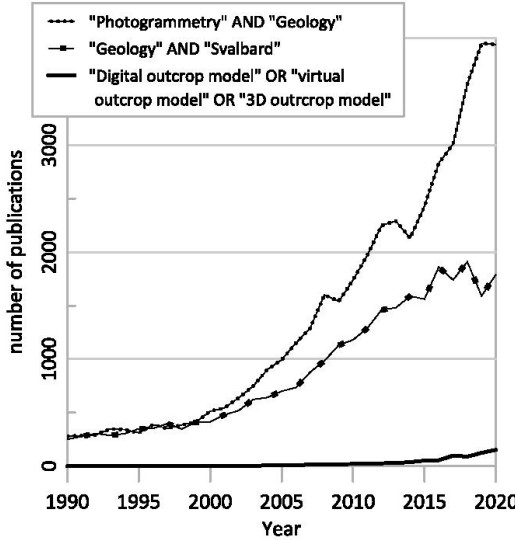




**Figure 2**: Overview of the Svalbox concept and its main elements. A) Screenshot from the UNIS-internal part of Svalbox, illustrating the correlation of multi-scale sedimentological logs from the Festningen outcrop integrated within the Petrel platform. B) Screenshot of the open-access part of Svalbox, with geological maps overlain with digital outcrop models, 360° imagery and geophysical data sets. Refer to Table 1 for details.

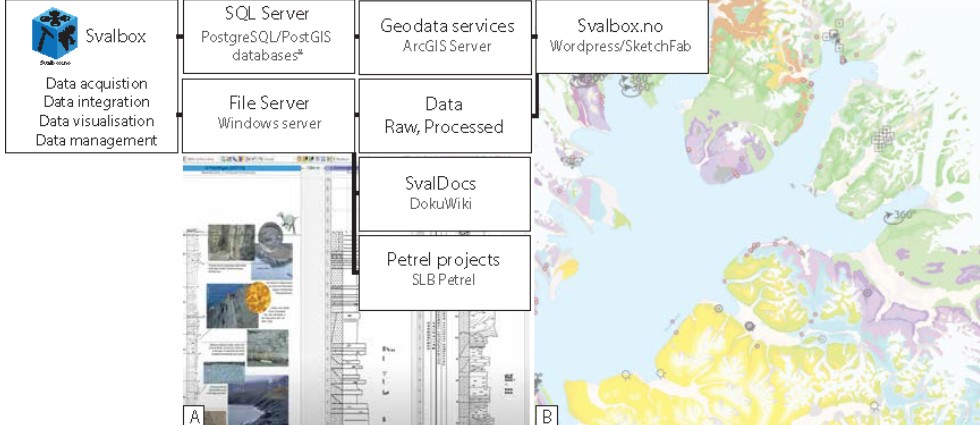



**Figure 3**: Location of Svalbard and the Billefjorden Trough, the main field area for the AG222 course. Winter
and summer conditions of the same mountain, Løvehovden (see http://toposvalbard.npolar.no for exact location)
are shown for comparison.

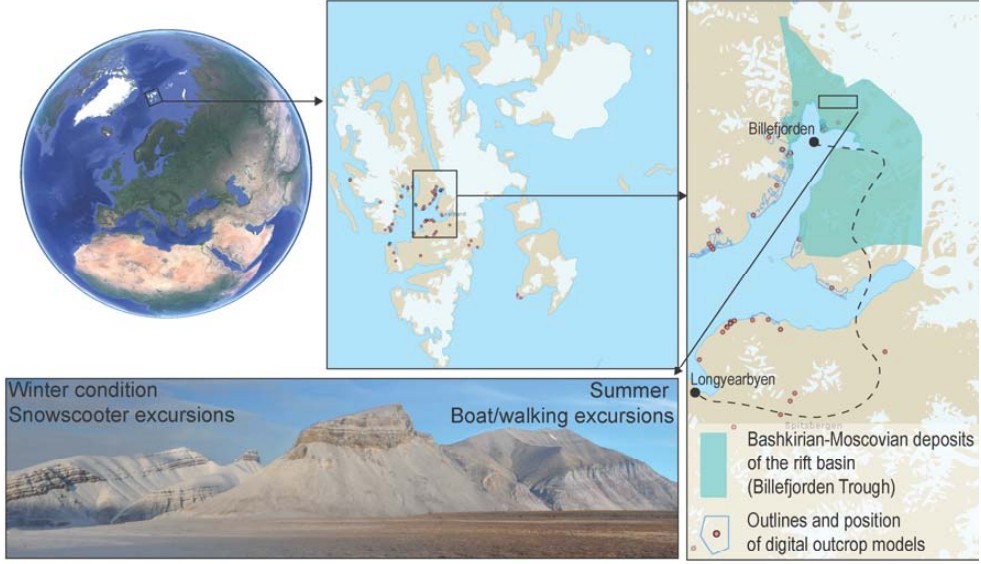





**Figure 4**: Extreme seasonal cycle, as exemplified by the amount of daylight hours, temperature, windspeed and snow depth in Adventdalen near Longyearbyen from 2017 to 2021 (meteorological source: https://klimaservicesenter.no/). The AG222 course and field periods are marked - these are characterized with maximum snow cover and lowest temperatures, and a rapid shift from no daylight to permanent daylight during the course period.

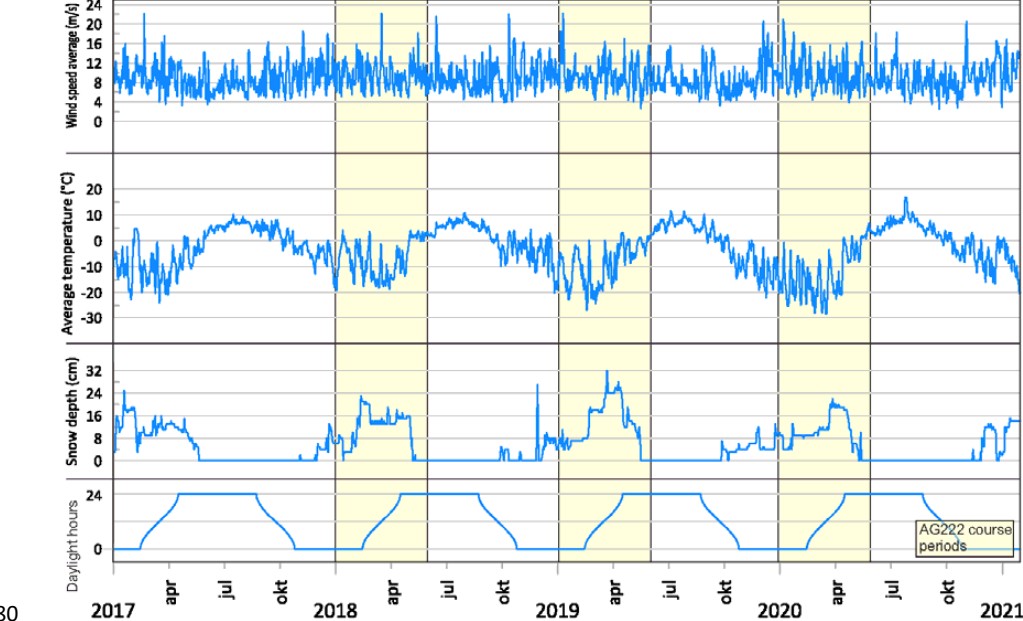






**Figure 5**: Overview of the central elements in the AG222 course, see Table 2 and 3 for details. The four course
modules are organised to build on each other, with skills learnt in the first half of the course highly relevant for
the second half. The two main graded group assessments are very intensive within a relatively short period,
while the graded individual assessment spans the entire semester giving student's flexibility to manage their
time. Field campaigns naturally follow the season, and each campaign has a clear learning objective. BH =
Botneheia day field trip, NT = near-town, PS = Polarsyssel (boat excursion on Isfjorden).






**Figure 6**: Screenshots from interpretation and integration of digital outcrop models, including textured DOM, slope calculation, elevation display and integration of the DOM with regional terrain models and geological

maps. The illustrated example is from Mediumfjellet (https://v3geo.com/model/142; details on the structural geology in Larsen (2010) and Strand (2015). A) Regional digital elevation model and geological map (both courtesy of Norwegian Polar Institute) overlain by Mediumfjellet DOM. B) Close up of digital outcrop model, with interpretations by Strand (2015). C) View of DOM from the south-east, with colour-coding representing the slope angle.

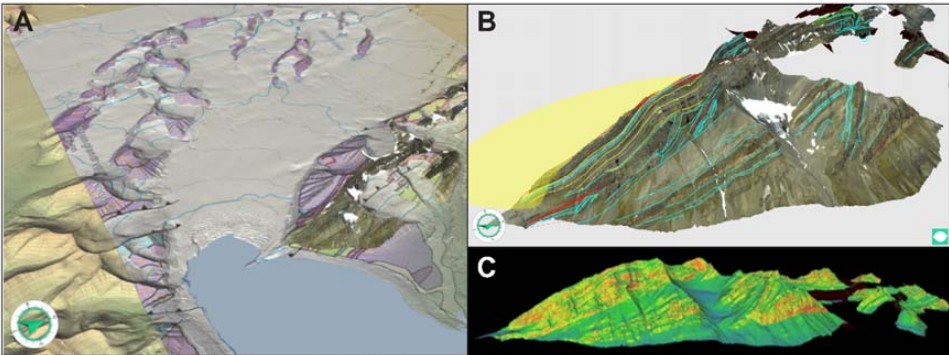




**Figure 7**: Examples from seismic modelling of the Kvalpynten digital outcrop model
(https://v3geo.com/model/90) conducted by the 2019 AG222 class on a ca. 2 km long part of the outcrop. A)

Interpretation of digital outcrop model using LIME. B) Assignment of elastic parameters to specific lithologies.
C) Seismic modelling under varying dominant frequencies using SeisRoX. D) Direct overlay of the seismic
model on the DOM.

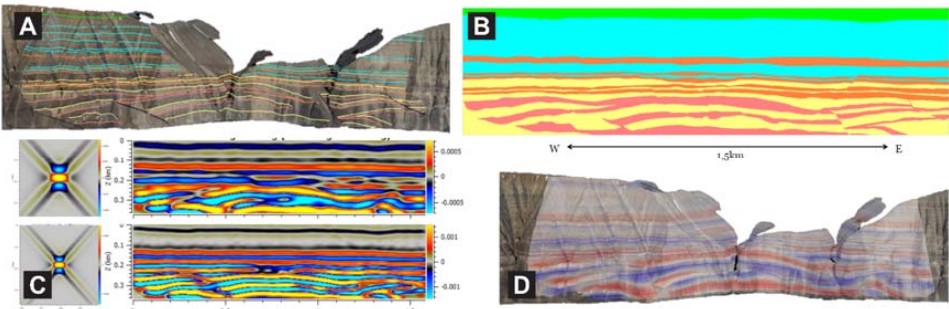




**Figure 8**: Examples from the Billefjorden license claim assignment, conducted by the 2019 AG222 class. A)
Top Reservoir structure contour map with fault zones and the area to be fictively claimed (red rectangle). B)
Well prognosis, illustrating key petroleum system elements like source and reservoir rocks. C) East-west cross
section across the proposed drill site.

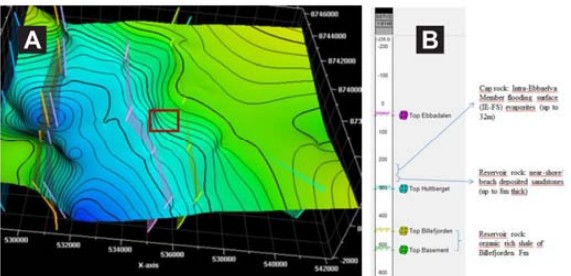

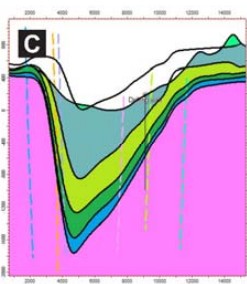






**Figure 9**: Applications of DOMs for concrete usage in teaching.

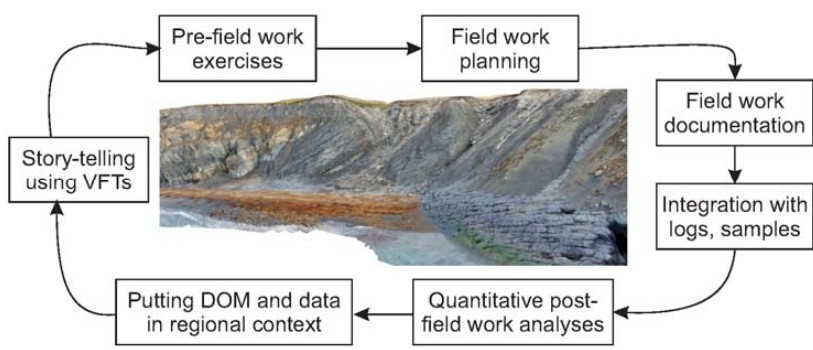






**Figure 10**: Schematic diagram of the synergies between the AG222 course (and other UNIS courses and research activities) and the Svalbox portal.

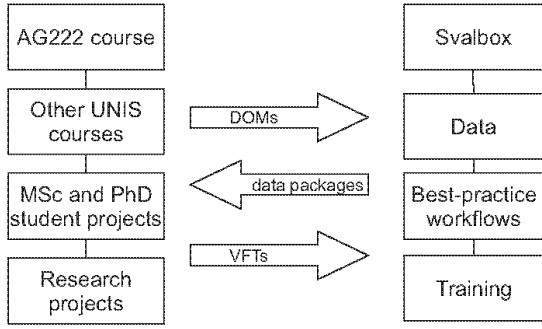




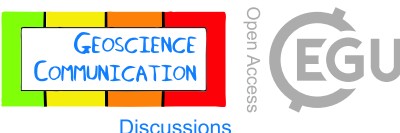

**Figure 11**: Consequences of the Covid-19 pandemic on the AG222 assessment related to the license claim application.

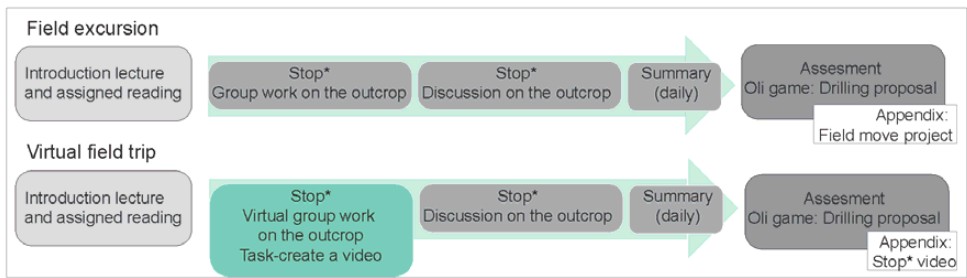