# Peer review of "Teaching with digital geology in the high Arctic: opportunities and challenges"

_Geoscience Communication, 2021_

## Referee Comment (RC1)

**Sizing Opportunities – give a handle, please.**

In the paper *Teaching with digital geology in the high Arctic: opportunities and challenges,* Kim Senger and co-worker report a fascinating experience. The text reads very well, and the supplementary material (online) is rich; also, the supporting bibliography is extensive. As it stands, the paper is publishable – as reporting experiences. However, it is a choice whether the paper's (technical) detail is suitable given a critical limitation regarding the reproducibility of the experience.

The authors communicate how modern tools (such as Digital Outcrop Models and Virtual Field Trips) can be used to teach Bachelor students about Svalbard's geology, to horn some technical skills (e.g. data management and software integration) of the students, and to educate them to prepare (and report from) geological fieldwork. The overall account is optimistic but not overblown. The joke ('rattlesnake' in line 312) is charming. However, the noticeable preference for the word 'exponentially' (line 80, 89. 323, 422) should be scaled down – using a logarithmic scale in Fig.1 likely show that the increase is not exponential.

The authors rightly stress that the particular requirements at Svalbard caused the early development of a set of tools and practices that are of much broader applicability, now as the COVID-19 pandemic forces to alter (traditional) teaching modes in favour of remote modes.

The 'open access' to a significant part of the course material will allow many lecturers, students and 'aficionados' to learn about Svalbard's fascinating geology. That is an additional strength of the paper. However, the buck stops there. The article lacks vital information to allow other teams to build similar tools (for their preferred location). Hence, the paper lacks the necessary information to reproduce how to teach and educate using modern communication tools.

The material gathered in the paper is impressive. However, the reader misses part of the 'methodology section', for example, the information about necessary preconditions for success (e.g. lasting cooperation with mining companies, public and private funding, skill-full individuals, limited legal concerns about privacy or access). Such information is essential to allow other institutions to set up similar schemes. Likewise, to learn about insights into probable causes of failures would be helpful;

this, as well for technology choices, supporting (IT)-infrastructure, advisable management structures, or required interpersonal skills.

Hence, teaching Svalbard's geology may cope with some 'shocks of the COVID-19 pandemic', as the experiences of the authors show. However, to teach the 'know-how' to cope with such shocks needs more than to report about events (= reporting observations). To illustrate the perceived lacuna, when seen from an educational / communication perspective: the paper shows an impressive 'educational outcrop' but does not analyse it, or the paper shows findings of an outcrop model but does not share the model code.

Drawing on the above, it is advisable to enrich the paper by reporting about 'preconditions for success & risks to fail' (before line 290) and discussing these preconditions (before line 410). Such a minor amendment seems mandatory for the benefit of the profession (and the reader); also, it would justify publishing the given detail. Finally, it would be 'nice to have' that the authors reflect a little about further opportunities of their experiences, e.g. for more open and participatory education, content accessible for anybody, and, tentatively, having a comprehensive outreach to non-professional communities.

---

## Author Response (AR1)

Dear Editor and Reviewers

Thank you very much for your constructive feedback. We have now revised the manuscript and provide a point-by-point response. For ease of reading our response is highlighted in red.

All the best,

Kim Senger (on behalf of all the authors)

Reviewer 1 comments:

**Sizing Opportunities – give a handle, please.**

In the paper Teaching with digital geology in the high Arctic: opportunities and challenges, Kim Senger and co-worker report a fascinating experience. The text reads very well, and the supplementary material (online) is rich; also, the supporting bibliography is extensive. As it stands, the paper is publishable – as reporting experiences. However, it is a choice whether the paper's (technical) detail is suitable given a critical limitation regarding the reproducibility of the experience.

Thank you for the input. We hope this work will be an inspiration for other groups seeking to obtain the same goals as us. We purposefully do not include "cookbook recipes" on how to manage such complex projects such as Svalbox/AG222 course, as different research/teaching groups will undoubtedly start on a different level with respect to financing (amount and longevity), geological access, licensing issues for non-open source software and previous experience. However, we have tried to revise the manuscript in a way to make our experiences from developing AG222/Svalbox even clearer, and thus more applicable elsewhere.

The authors communicate how modern tools (such as Digital Outcrop Models and Virtual Field Trips) can be used to teach Bachelor students about Svalbard's geology, to horn some technical skills (e.g. data management and software integration) of the students, and to educate them to prepare (and report from) geological fieldwork. The overall account is optimistic but not overblown. The joke ('rattlesnake' in line 312) is charming.

Thank you for the input! Just a comment – a number of the authors encountered rattlesnakes while on fieldwork in Utah which were certainly not charming 😊

However, the noticeable preference for the word 'exponentially' (line 80, 89. 323, 422) should be scaled down – using a logarithmic scale in Fig.1 likely show that the increase is not exponential.

Changed "exponentially" to "rapidly" in places, removed elsewhere.

The authors rightly stress that the particular requirements at Svalbard caused the early development of a set of tools and practices that are of much broader applicability, now as the COVID-19 pandemic forces to alter (traditional) teaching modes in favour of remote modes.
The 'open access' to a significant part of the course material will allow many lecturers, students and 'aficionados' to learn about Svalbard's fascinating geology. That is an additional strength of the paper. However, the buck stops there. The article lacks vital information to allow other teams to build similar tools (for their preferred location). Hence, the paper lacks the necessary information to reproduce how to teach and educate using modern communication tools.

The material gathered in the paper is impressive. However, the reader misses part of the 'methodology section', for example, the information about necessary preconditions for success (e.g. lasting cooperation with mining companies, public and private funding, skill-full individuals, limited legal concerns about privacy or access). Such information is essential to allow other institutions to set up similar schemes. Likewise, to learn about insights into probable causes of failures would be helpful; this, as well for technology choices, supporting (IT)-infrastructure, advisable management structures, or required interpersonal skills.

Hence, teaching Svalbard's geology may cope with some 'shocks of the COVID-19 pandemic', as the experiences of the authors show. However, to teach the 'know-how' to cope with such shocks needs more than to report about events (= reporting observations). To illustrate the perceived lacuna, when seen from an educational / communication perspective: the paper shows an impressive 'educational outcrop' but does not analyse it, or the paper shows findings of an outcrop model but does not share the model code.

Drawing on the above, it is advisable to enrich the paper by reporting about 'preconditions for success & risks to fail' (before line 290) and discussing these preconditions (before line 410). Such a minor amendment seems mandatory for

the benefit of the profession (and the reader); also, it would justify publishing the given detail. Finally, it would be 'nice to have' that the authors reflect a little about further opportunities of their experiences, e.g. for more open and participatory education, content accessible for anybody, and, tentatively, having a comprehensive outreach to non-professional communities.

Added "Applications of Svalbox beyond Svalbard" section in discussion

The Svalbox concept can, in theory, be established also in other locations worldwide. The main requirements are access to high-quality outcrops with varied geological features, complementary surface and subsurface data, and mid- to long-term funding to allow not just development but also regular updates. Clearly, programming skills are required in the team, together with local geological knowledge and data management skills. Ideally, a Svalbox-type project has a full-time data manager who can conduct regular data updates and develop the project over a longer-term. At present, Svalbox is essentially run on the side of other teaching and research duties, with key expertise covered by temporary PhD student duty hours, which is considered the biggest risk in terms of the project's sustainability over the long term. Nonetheless, the current focus on digitization across the society, access to open access data and focus on emerging technologies, promises opportunities to seek longer-term funding also with longer-term staffing.

A large part of Svalbox' success relates to the direct linkage with the AG-222 course and its students, who still represent the main user group of the portal. The students of AG-222 have a varied background from different Norwegian, Nordic and international universities. The students are introduced to state-of-the-art technology and teaching methods in the BSc-level AG-222, which may inspire and influence their future career choices and skill sets for example when pursuing a MSc project. Ideally, the AG-222 students may also bring some new knowledge and skills back to their home institutions, and thus contribute to broadening and advancing this type of technology-based and research-oriented teaching beyond Svalbard.

Furthermore, the strength of Svalbox compared to other DOM repositories in the world lies in the direct integration of DOMs with other geoscientific data. As far as we are aware, there are no other databases with such a rich (and growing) data set that integrates both DOMs and other geoscientific data. The geological playground of Svalbard with its rich history of coal and hydrocarbon exploration (Senger et al., 2019) and the well-exposed and stratigraphically diverse outcrops, provides a unique opportunity to develop Svalbard into a digital teaching and training hub for use beyond UNIS. For instance, the introduction of Svalbard's geology or the Festningen section is already an element of various courses at Norwegian mainland universities. In addition, Svalbox is already a valuable tool for planning fieldwork for bachelor and master students doing data collection for their thesis projects, and for national and international researchers that regularly visit the archipelago.

Reviewer 2 comments:

Geosci. Commun. Discuss., referee comment RC2
https://doi.org/10.5194/gc-2021-6-RC2, 2021
**Comment on gc-2021-6**

Rachel Bosch (Referee)

Referee comment on "Teaching with digital geology in the high Arctic: opportunities and challenges" by Kim Senger et al., Geosci. Commun. Discuss.,
https://doi.org/10.5194/gc-2021-6-RC2, 2021

**Referee#2 comments on Senger et al.: Teaching with digital geology in the high Arctic: opportunities and challenges**

**Geoscience Communication instruction to referees:**

'Generally, a referee comment should be structured as follows: an initial paragraph or section evaluating the overall quality of the preprint ("general comments"), followed by a section addressing individual scientific questions/issues ("specific comments"), and by a compact listing of purely technical corrections at the very end ("technical corrections": typing errors, etc.).'

**General Comments**

This manuscript presents a unique undergraduate geology course curriculum, as the instructor(s) have been implementing digital field technologies for several years to account for extreme latitude conditions that make it hard to provide in-person field instruction during the winter. When the covid-19 pandemic forced instructors to pivot to teaching virtually, Senger et al. already had the structure in place to make this transition. Sharing these resources with the geoscience community is a great contribution and provides others with the opportunity to enhance their teaching. This manuscript is wellsuited for the Geoscience Communication special issue Virtual geoscience education resources upon revision.

Thank you for your kind comments

**Specific Comments**

If Virtual Reality is routinely used in the AG222 course, it should be introduced and its implementation described earlier in the manuscript. It seems out of place to bring additional technology into the manuscript in the Discussion in its whole own section.

VR paragraph is removed. While VR was used in some of the AG222 course elements over the years, it is not something that is consistently used at the moment. We do however envision a follow-up contribution highlighting the use of VR in the AG222 course in the near-future.

This manuscript seems entirely focused on the course AG222. I suggest eliminating the paragraph on AG-209 that begins on line 402. It does not add new or different information.

AG-209 paragraph is removed, and replaced with a short introduction to this "sister course" of AG-222 (line 165)

**Technical Corrections**

Three of the figures were not referred to in the text of the manuscript: Fig. 2, Fig. 5, and Fig. 7. Add references where appropriate or eliminate those figures if they are not relevant to the text.

In-text references added for Fig. 2, 5 and 7 in relevant sections.

Please review your Tables and Table numbering. The tables are referred to in this order in the text: Table 2 (line 156), Table 3 (line 161), and Table 1 (line 414). Tables 4 and 5 are not referred to in the text. Electronic Appendix Supplementary Table 1 and Electronic Appendix Supplementary Table 2 are confusing names since they reuse the terms Table 1 and Table 2, respectively. If they are appendices, consider renaming them A and B instead of 1 and 2? These two supplemental tables are also not referred to in the text. As with the figures, please add references where appropriate or eliminate those tables if they are not relevant to the text.

Table 1 is now referred to earlier in text
Reference Table 4 and 5 now added in the text
Supplementary material tables renamed as Table A and Table B

Throughout: When using the abbreviation "e.g.," it is always followed by a comma (e.g., like this).

Ok, fixed

Throughout: When citing multiple sources parenthetically, insert a space after each semicolon.

Ok, fixed

In the section, "Future perspectives," you state that you are building on FAIR data principles (line 421). On line 214, you write you "have adopted the online ArcGIS StoryMaps approach." As far as I am aware, this is commercially licensed software. Have you considered an open-source approach such as using Google Earth Projects?

Very valid point – we have clarified this by adding the following comments in line 214: "StoryMaps is a commercial product, but a site license is available at UNIS for its unlimited usage."

And line 421: "This approach applies not just to the data sets, but also tools to be used. Both StoryMaps and Petrel are licensed software, though with academic rates (Petrel is, for instance, at the time being available for free for academic institutions like UNIS). In the future, we envision testing and potentially adopting open-source solutions for the entire Svalbox value chain."

Line 23: "spectacular" is subjective and unscientific—replace with a more objective, descriptive adjective.

Ok, changed to "high-quality"

Line 28: Please rephrase the first sentence so it better conveys your meaning.
Changed to "DOMs allow to bring geoscientists to the outcrops digitally, which is particularly important in view of the Covid-19 pandemic that restricts travel and thus direct access to outcrops."

Line 54: LiDAR is an acronym and needs to be defined on first usage. On line 54, it was spelled, "lidar," and on line 514, it was spelled, "Lidar."
Ok, fixed

Line 57: Change "as" to "has".
Ok, fixed

Line 135: Change "that" to "who".
Ok, fixed

Line 149: I suggest you change "component" to "emphasis".
Ok, fixed

Line 177: Change "own image" to "own-image".
Changed to "image acquisition of outcrops by the students"

Line 195: Add a comma after "imagery".
Ok, fixed

Line 222: "an excellent" is unnecessarily subjective—recommend changing it to a more objective, descriptive adjective (why is it excellent?), or remove and replace with "a".
Ok, changed to "a very-well exposed and well-studied case study"

Line 254: "company geologist" should not be hyphenated.
Ok, fixed

Line 266: Change "allow" to "allows".
Ok, fixed

Line 302: Regarding the software examples in parenthesis, please choose either the word, "or," or the comma and use it in both sets of examples.
Ok, fixed

Line 373: The sentence that begins "Although geological guides . . ." needs to be reworded for parallel structure. Here is a suggestion, "Although geological guides are handed out, presentations are given in the evening of what to see the next day and are repeated the next morning, it is. . . ."

Ok, fixed

Line 402: Should the hyphen be removed from "AG-209" for consistency with "AG222"?
Ok, fixed (AG-209 paragraph removed)